# The feasibility of soluble Fms-Like Tyrosine kinase-1 (sFLT-1) and Placental Growth Factor (PlGF) ratio biomarker in predicting preeclampsia and adverse pregnancy outcomes among medium to high risk mothers in Kuala Lumpur, Malaysia

Nurul Afzan Aminuddin[1]☯, Rosnah Sutan[1]☯*, Zaleha Abdullah Mahdy[2]☯, Rahana Abd Rahman[2]☯, Dian Nasriana Nasuruddin[3]☯

1 Community Health Department, Faculty of Medicine, Universiti Kebangsaan Malaysia, Kuala Lumpur, Malaysia, 2 Obstetrics and Gynaecology Department, Faculty of Medicine, Universiti Kebangsaan Malaysia Medical Center, Kuala Lumpur, Malaysia, 3 Pathology Department, Faculty of Medicine, Universiti Kebangsaan Malaysia, Kuala Lumpur, Malaysia

☯ These authors contributed equally to this work.

* rosnah_sutan@yahoo.com

## Abstract

### Background

Preeclampsia significantly contributes to maternal and perinatal morbidity and mortality. It is imperative to identify women at risk of developing preeclampsia in the effort to prevent adverse pregnancy outcomes through early intervention. Soluble fms-like tyrosine kinase-1 (sFlt-1) and placental growth factor (PlGF) level changes are noticeable several weeks before the onset of preeclampsia and its related complications. This study evaluated the feasibility of the sFlt-1/PlGF biomarker ratio in predicting preeclampsia and adverse pregnancy outcomes using a single cut-off point of >38.

### Methods

This is a prospective cohort study conducted at a single tertiary centre, in an urban setting in Kuala Lumpur, Malaysia, between December 2019 and April 2021. A total of 140 medium to high risk mothers with singleton pregnancies were recruited at ≥20 weeks' gestation. sFlt-1/PlGF ratio was measured and the participant monitored according to a research algorithm until delivery. The primary outcome measure was incidence of preeclampsia and the secondary outcome measure was incidence of other adverse pregnancy outcomes.

### Results

The overall incidence of preeclampsia was 20.7% (29/140). The mean sFlt-1/PlGF ratio was significantly higher in preeclampsia (73.58 ± 93.49) compared to no preeclampsia (13.41 ±

**Data Availability Statement:** All relevant data are within the paper and its Supporting Information files.

**Funding:** This research received funding through the Fundamental Grant (FF-2019-371) and Matching Grant (FF-2019-371/1) from Universiti Kebangsaan Malaysia. Provision of Elecsys sFlt-1: PlGF and PreciControl Multimaker test kits from Roche Diagnostics International Ltd is acknowledged. The funders had no role in study design, data collection and analysis, decision to publish, or preparation of the manuscript.

**Competing interests:** Funding This research received funding through the Fundamental Grant (FF-2019-371) and Matching Grant (FF-2019-371/ 1) from Universiti Kebangsaan Malaysia. Conflict of Interest Provision of Elecsys sFlt-1/PlGF and PreciControl Multimaker test kits from Roche Diagnostics International Ltd is acknowledged. This does not alter our adherence to PLOS ONE policies on sharing data and materials.

21.63) (p = 0.002). The risk of preeclampsia (adjusted OR 28.996; 95% CI 7.920–106.164; p<0.001) and low Apgar score (adjusted OR 17.387; 95% CI 3.069–98.517; p = 0.028) were significantly higher among women with sFlt-1/PlGF ratio >38 compared with sFLT-1/PlGF ratio ≤38. The area under the receiver-operator characteristic curve (AUC) for a combined approach (maternal clinical characteristics and biomarker) was 86.9% (p<0.001, 95% CI 78.7–95.0) compared with AUC biomarker alone, which was 74.8% (p<0.001, 95% CI 63.3–86.3) in predicting preeclampsia. The test sensitivity(SEN) was 58.6%, specificity (SPEC) 91%,positive predictive value (PPV) 63% and negative predictive value (NPV) 89.3% for prediction of preeclampsia. For predicting a low Apgar score at 5 minutes, the SEN was 84.6%, SPEC 87.4%, PPV 40.7%, and NPV 98.2%; low birth weight with SEN 52.6%,SPEC 86.0%, PPV 37.0%, NPV 92.0%; premature delivery with SEN 48.5%, SPEC 89.5%, PPV 59.3%, NPV 84.7% and NICU admission with SEN 50.0%, SPEC 85.8%, PPV 37.0% and NPV 91.2%.

## Conclusions

It is feasible to use single cut-off point of >38 ratio of the biomarkers sFlt-1/PlGF in combination with other parameters (maternal clinical characteristics) in predicting preeclampsia and adverse pregnancy outcomes among medium to high risk mothers without restricting outcome measurement period to 1 and 4 weeks in a single urban tertiary centre in Kuala Lumpur, Malaysia.

## Introduction

Preeclampsia complicates 2–10% of pregnancies worldwide and significantly contributes to maternal, fetal, and newborn morbidity and mortality [1–5]. Preeclampsia is characterized by the combined presentation of hypertension with proteinuria and/or maternal organ dysfunction, such as renal insufficiency, liver involvement, neurological or haematological complications, and uteroplacental dysfunction as evidenced by fetal growth restriction (FGR) [2, 6].

Early detection and prevention of preeclampsia will reduce both prevalence of the disease and health service costs [7]. Early identification of women at risk of developing preeclampsia for commencement of aspirin and calcium as prophylaxis, close monitoring, and timely delivery, are critical steps in the management, in order to prevent adverse pregnancy outcomes [8, 9]. A huge challenge in managing preeclampsia is in predicting its development and severity [10].

The pathophysiology of preeclampsia remains unclear but placental dysfunction is believed to be the primary insult [11, 12]. Incomplete maternal spiral artery remodeling in early pregnancy leads to placental hypoperfusion. Oxidative stress within the placenta leads to changes in biomarker levels, with increased serum soluble Fms-like tyrosine kinase-1 (sFlt-1) and reduced placental growth factor (PlGF) [13].

PlGF is a proangiogenic factor [14] that has been found to be a biomarker of preeclampsia [15, 16] It is produced by the placenta, mainly the syncytiotrophoblast, and the endothelium [16, 17]. PlGF plays a major role in the growth of placental blood vessels [18, 19]. It can be detected in maternal blood from 8 weeks' gestation up to the second trimester of pregnancy, after which PlGF decreases progressively until delivery [20]. In preeclampsia, PlGF remains persistently low [16, 21, 22]. Infusion of recombinant human PlGF using intraperitoneal

osmotic mini-pumps can eliminate the progression of hypertension in a rat model of pre-eclampsia [23].

Vascular endothelial growth factor (VEGF) is important for vascular homeostasis which stimulates both VEGF receptor-1 (VEGFR-1) and -2 (VEGFR-2), VEGF and endothelial nitric oxide synthase (eNOS), are essential for angiogenesis [24–28]. sFlt-1 antagonises VEGF and PlGF, preventing interaction of VEGF and PlGF, by preventing interaction with the endothelial receptors and causing endothelial dysfunction [29–31].

Changes in the levels of sFlt-1 and PlGF are noticeable several weeks before the onset of pre-eclampsia and its related complications [11]. sFlt-1/PlGF ratios of 38 or lower carry a negative predictive value (NPV) of 99.3% (95% CI 97.9–99.9) in the subsequent week, with 80.0% sensitivity (95% CI 51.9–95.7) and 78.3% specificity (95% CI 74.6–81.7) [32], whereas ratios above 38 carry a positive predictive value (PPV) of 36.7% (95% CI 28.4–45.7) within 4 weeks, with 66.2% sensitivity (95% CI 54.0–77.0) and 83.1% specificity (95% CI 79.4–86.3) [32]. In the Malaysian clinical practice guidelines (CPG), current preeclampsia screening relies on clinical characteristics and proteinuria only [33], which are of poor predictive value [34]. At present, women with suspected preeclampsia are unnecessarily admitted to hospital [35, 36], whilst asymptomatic severe cases are overlooked and managed as outpatient, resulting in late presentation with irreversible complications [35].

Serum sFlt-1/PlGF ratio testing (from 20 weeks' gestation onwards) has been suggested by the latest Malaysia CPG in order to predict preeclampsia [33], however it is not widely implemented especially in government health clinics setting, despite widespread adoption in several countries [35]. The National Institute for Health and Care Excellence (NICE), United Kingdom (UK), has recommended sFlt-1/PlGF ratio ≤38 to be used as a rule-out test for suspected pre-eclampsia between 20 and 34 weeks' gestation [37]. The German Society of Gynecology and Obstetrics (DGGG) recommended use of second trimester sFLT-1/PlGF ratio with uterine artery Doppler for risk assessment of preeclampsia development and prognostic assessment [38].

These biomarkers may help improve early diagnosis of preeclampsia and enable prediction of maternal and fetal outcomes [11]. We set out to assess the feasibility of sFlt-1/PlGF ratio in predicting preeclampsia and adverse pregnancy outcomes using the recently proposed single cut-off values of 38 and preeclampsia screening algorithm developed.

## Materials and methods

A prospective cohort study was conducted at a single tertiary centre in an urban setting in Kuala Lumpur, Malaysia, from December 2019 until April 2021. The aim was to evaluate the feasibility of using sFlt-1/PlGF immunoassay ratio for early identification of preeclampsia and related adverse pregnancy outcomes. The study participants were monitored based on the algorithm shown in Fig 1 from enrolment into the study at ≥20 weeks' gestation until delivery. If the sFLT-1/PlGF ratio test was negative (≤38), the blood test will be repeated weekly for continued monitoring. If the result was positive (>38), the test will be repeated four-weekly until delivery or development of preeclampsia. Maximum 4 sample of sFlt-1/PlGF ratio test each participant for the purposed of controlling research cost. The outcome measure was pre-eclampsia and related adverse pregnancy outcomes as a comparison between positive and negative tests.

The minimum sample size was calculated based on the formula for prospective cohort studies by Fleiss (1981) [39], with a power of 80% and confidence interval (CI) of 95% [40]. A minimum of 120 samples were required.

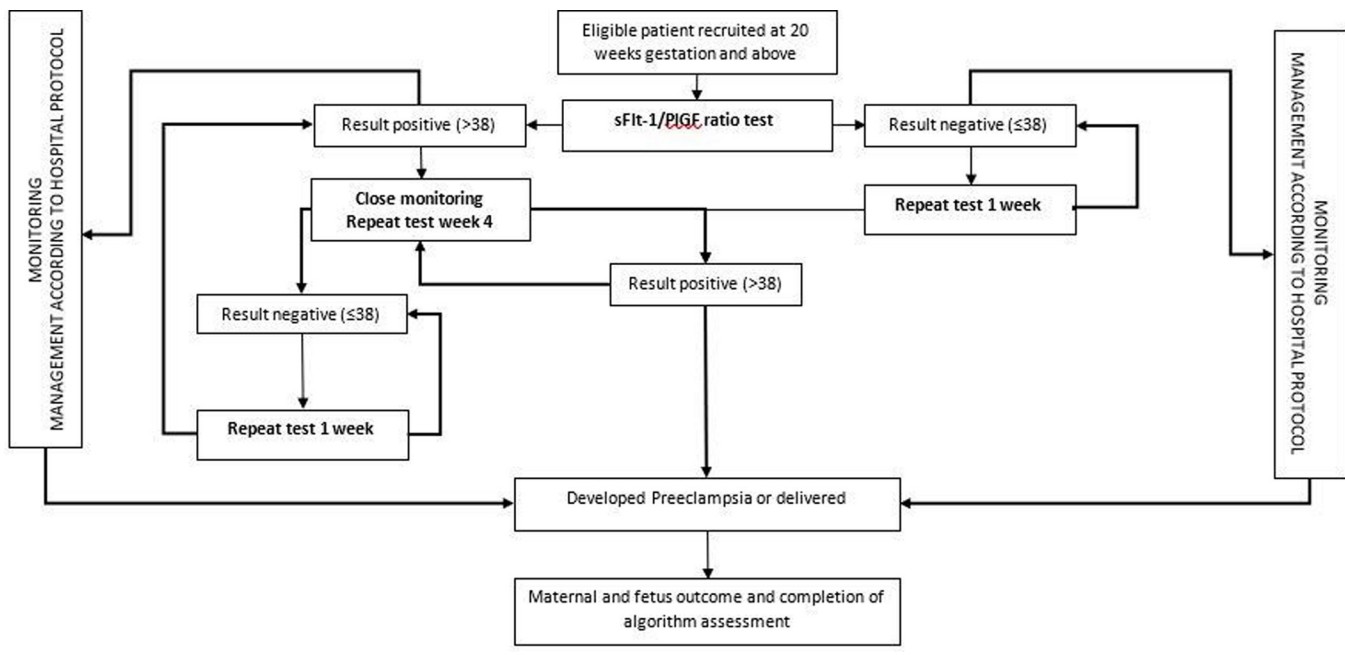

**Fig 1. Preeclampsia screening algorithm.**

We included women aged ≥18 years, with systolic blood pressure (SBP) of 140–160 mmHg or diastolic blood pressure (DBP) of 90–100 mmHg or gestational hypertension or underlying chronic hypertension, and singleton pregnancies of ≥20 weeks gestation. We excluded women with confirmed diagnosis of preeclampsia or eclampsia, fetal congenital malformation, and those who refused.

The study received ethical approval from the Ministry of Health Malaysia (NMRR-19-1104-46049(IIR)) and UKM (FF-2019-371). The study conducted according to the Declaration of Helsinki. Written informed consents were obtained from all participants.

Whole blood was collected in a 5 ml plain vacutainer tube on the recruitment day. The blood was centrifuged, then subjected to a calibrated **cobas e** 411 immunoassay analyser tests (Roche Diagnostics). Quality control was performed for each test by using PRECICONTROL multimarker reagent. The blood analysis was performed by a trained medical laboratory technician under the supervision of a chemical pathologist. The laboratory result was entered in an integrated laboratory management system (ILMS) linked to the hospital information system.

## Primary outcome measure

Our primary outcome measure was preeclampsia as defined by the International Society for the Study of Hypertension in Pregnancy (ISSHP) [2], i.e. gestational hypertension accompanied by at least one of the following new onset conditions at or after 20 weeks' gestation: proteinuria and/or maternal organ dysfunction (including acute kidney injury, altered liver function, neurological complications, haematological complications or uteroplacental dysfunction).

## Secondary outcomes measure

Maternal and fetal adverse pregnancy outcome including maternal death, maternal intubation, emergency caesarean section, disseminated intravascular coagulation (DIVC), placental

abruption, acute heart failure, low birth weight, neonatal death, intrauterine death, neonatal intensive care unit (NICU) admission, premature delivery, and low Apgar score. The percentages of completed the preeclampsia screening algorithm developed is also measured as secondary outcome (as shown in Fig 1).

The operational definition of maternal death is death of a woman during pregnancy or up to 42 days post-partum. Maternal intubation is defined as the need for maternal intubation as a result of preeclampsia and related complications. Emergency caesarean section is defined as emergency caesarean section with preeclampsia as the main indication. Disseminated intravascular coagulation (DIVC), abruptio placenta, and acute heart failure are defined as complications resulting directly from preeclampsia. A low-birth-weight infant are defined as baby's birth weight of <2.5 kg. Intrauterine death is defined as death of a fetus in utero after 22 weeks of gestation. Neonatal death is defined as death of a baby within the first 28 days of life. Neonatal intensive care unit (NICU) admission is defined as requirement for NICU admission after birth. Premature delivery is defined as birth <37 weeks of gestation. A low Apgar score is defined as Apgar score <7 at 5 minutes after birth.

## Independent variables

Our independent variables comprise sociodemographic characteristics (age, race, occupation, household income, education level), clinical features, Body mass index (BMI) classification, antenatal risk, gravida classification, parity classification, gestation at recruitment, anaemia, pre-gestational diabetes mellitus (DM), gestational DM, renal disease, autoimmune disease, previous history of hypertensive disorders in pregnancy, chronic hypertension, and sFlt-1/PlGF ratio.

Age was defined in years at recruitment based on date of birth, and categorized into two groups (>35 years, ≤ 35 years). Race was categorized into Malay and non-Malay according to major ethnicities in Malaysia. Occupation was classified according to International Standard Classification of Occupations and further divided into professional and non-professional groups. Household income was defined as overall income that was earned by household members, whether in cash or kind, and can be referred to as gross income. Household income was categorized into three groups Top 20 (> RM 10 960), Middle 40 (RM 4850–10959) and Below 40 (<RM4849). Education level was categorised into two groups (tertiary, secondary).

Body mass index (BMI) calculated as body weight in kg divided by height in meter squared (kg/m2), classified according to international World Health Organization (WHO) BMI classifications and further categorized into two groups (BMI ≥25, BMI ≤24.9). Antenatal risk was categorized into yellow (high risk) and green (medium risk), according to the Malaysian antenatal risk assessment colour coding system [41]. Gravida was categorized into three groups (Gravida 1, 2–5, ≥6). Parity was categorized into three groups (Para 0, 1–4, ≥5). Gestation at recruitment was defined as period of gestation (in weeks) at recruitment, either based on the date of the last menstrual period or on ultrasonography. The gestation at recruitment was categorized into two groups (≥27 weeks, 20–26 weeks). Diseases in pregnancy i.e. anaemia, pregestational and gestational DM, renal disease, autoimmune disease, history of hypertensive disorder in pregnancy, and chronic hypertension were dichotomously categorized (yes, no). sFlt-1/PlGF ratio was also dichotomously categorized (positive >38, negative ≤38).

## Statistical analysis

Data were analysed using the IBM Statistical Package for the Social Sciences (SPSS) version 22. Incidence of preeclampsia was calculated from the number of preeclampsia cases divided by the total recruited sample. The characteristics of the variables are described using frequency

(n) and percentage (%). Simple logistic regression and Pearson Chi-square was used for bivariate analysis and further analysed using multiple logistic regression to control for potential confounders. All $p$-values resulted from two-sided statistical tests. Values of $p < 0.05$ were considered statistically significant.

The diagnostic validity for sFlt-1/PlGF ratio was calculated and presented as sensitivity, specificity, PPV, NPV, positive likelihood ratio, negative likelihood ratio and post-test probability. The receiver operator characteristic (ROC) curves were calculated and the area under the curves (AUCs) were reported.

## Result

### Study population

A total of 140 women were recruited between December 2019 and April 2021 as shown in Fig 2. The numbers of potentially eligible were 200. After being examined for eligibility, only 180 women were confirmed eligible. However, 40 women refused to involve in the study due to other life commitments. Only 57% (n = 80) completed the research algorithm. A total of 140 women were included in the full analysis population according to first blood sample taken.

Incidence of preeclampsia among medium to high risk mothers was 20.7% (29/140). The mean sFlt-1/PlGF ratio was significantly higher in women who developed preeclampsia (73.58 ± 93.49) compared with those who did not (13.41 ± 21.63), p = 0.002. The percentages of women who developed preeclampsia according to sFlt-1/PlGF ratio versus week of assessment are shown in **Table 1.** A total of 5 (62.5%) women developed preeclampsia within a week of positive sFlt-1/PlGF ratio (>38). At the first hospital booking and in the first week after initial blood screening, 29.4% and 23.5% of cases were screened positive and developed preeclampsia respectively. The percentage of patient developed preeclampsia among positive sFlt-1/PlGF ratio test subsequently reduced from week to week.

### Predictors of preeclampsia

**Table 2** shows factors associated with preeclampsia as defined by the ISSHP [2]. Simple logistic regression analysis indicated that occupation and positive sFlt-1/PlGF ratio (>38) at first blood sample are significantly associated with preeclampsia. On performing multivariate analyses using multiple logistic regression, adjusted analysis showed that women working as professionals with positive sFlt-1/PlGF ratio had a higher risk of developing preeclampsia compared to non-professionals and negative sFlt-1/PlGF ratio respectively.

### The association between positive sFlt-1/PlGF ratio and adverse pregnancy outcome

**Table 3** shows the association between positive sFlt-1/PlGF ratio test and adverse pregnancy outcome. After controlling with potential confounders including maternal characteristics, antenatal risk, underlying medical illness, gestational age and preeclampsia, adjusted analysis showed that the risk of low Apgar score (<7) at 5 minutes, low birth weight, premature delivery and baby admitted to NICU are significantly higher among positive sFlt-1/PlGF ratio test (>38) as compared to negative sFlt-1/PlGF ratio test (≤38).

### The validity of sFlt-1/PlGF ratio in predicting preeclampsia and low Apgar score at 5 minutes

**Table 4** shows the validity assessment of positive sFlt-1/PlGF ratio in predicting preeclampsia and adverse pregnancy outcome. The sensitivity of the test was low (58.6%) but the specificity

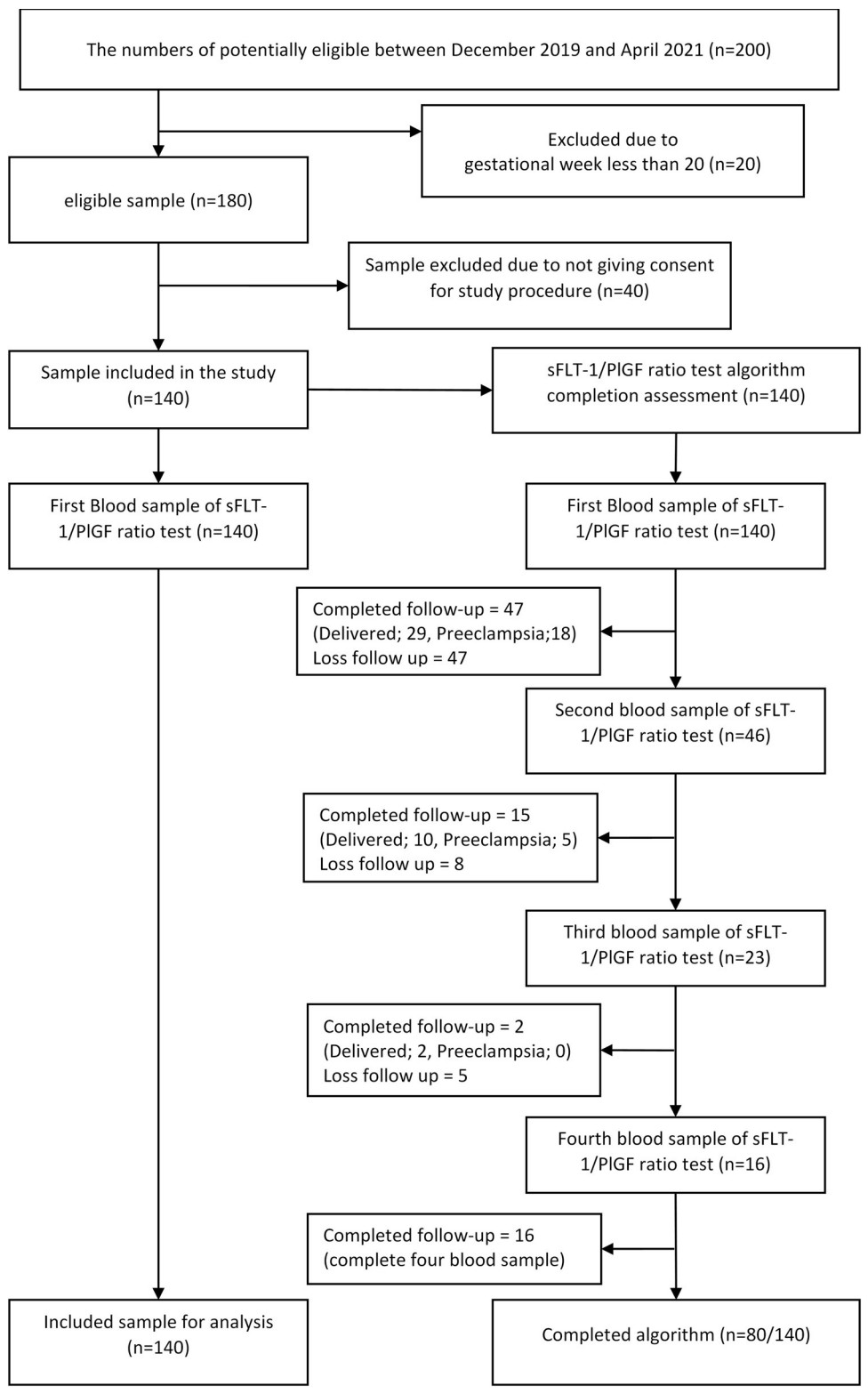

**Fig 2. Study participants flow chart.**

**Table 1. The number of cases screened as preeclampsia by week of assessment.**

| Assessment week | sFlt-1/PlGF ratio test | | Total |
| | Positive (>38) | Negative (≤38) | |
| | n(%) | n(%) | |
|---|---|---|---|
| 0 | 5(62.5%) | 3(37.5%) | 8(27.6%) |
| 1 | 4(57.1%) | 3(42.9%) | 7(24.1%) |
| 2 | 3(100.0%) | 0(0.0%) | 3(10.3%) |
| 3 | 2(50.0%) | 2(50.0%) | 4(13.8%) |
| 4 | 2(100.0%) | 0(0.0%) | 2(6.9%) |
| 5 | 0(0.0%) | 2(100.0%) | 2(6.9%) |
| 6 | 0(0.0%) | 1(100.0%) | 1(3.4%) |
| 7 | 1(100.0%) | 0(0.0%) | 1(3.4%) |
| 8 | 0(0.0%) | 0(0.0%) | 0(0.0%) |
| 9 | 0(0.0%) | 1(100.0%) | 1(3.4%) |
| Total | 17 | 12 | 29(100.0%) |

was high (91.0%) in predicting preeclampsia. **Fig 3** shows the comparison of ROC curve combined approach (maternal characteristic and sFlt-1/PlGF ratio) and sFlt-1/PlGF ratio alone in predicting both preeclampsia and significant adverse pregnancy outcome (low Apgar score). The area under the ROC curve (AUC) for combined approach was 86.9% (p<0.001, 95% CI 78.7–95.0) compared with AUC for sFlt-1/PlGF ratio alone 74.8% (p<0.001, 95% CI 63.3–86.3) to predict preeclampsia, the test having high sensitivity (84.6%) and specificity (87.4%) to predict low Apgar score at 5 minutes. The AUC for combined approach was 87.8% (p<0.001, 95% CI 75.2–100.0) compared with AUC sFlt-1/PlGF ratio alone 86.0% (p<0.001, 95% CI 74.2–97.9) to predict low Apgar score at 5 minutes.

## Discussion

We found that the mean sFlt-1/PlGF ratio was significantly higher among women who developed preeclampsia compared to women without preeclampsia, when using sFlt-1/PlGF ratio cut-off value >38 (sensitivity 58.6%; specificity 91.1%; PPV 63%; NPV 89.3% and AUC 74.8%, p<0.001; 95% CI 63.3–86.3). With combined approach (by including maternal characteristics and clinical features) the AUC improved towards 86.9% (p<0.001, 95% CI 78.7–95.0) compared with AUC sFlt-1/PlGF ratio alone in predicting preeclampsia. Other findings suggest that sFlt-1/PlGF ratio with or without clinical characteristics can help second or third trimester prediction of both early onset and late onset preeclampsia [11]. The discrepancy of these findings may be due to the relatively small sample size in our study, whereas other studies reported that the combined approach had superior predictive value compared to biomarkers in the first trimester alone [42–44]. Therefore, our study supports the predictive value of sFlt-1/PlGF ratio >38 for preeclampsia among Malaysian women with significantly higher PPV (63%) than the reported PPV (32–36%) by PROGNOSIS and PROGNOSIS Asia Study [32, 45].

The PROGNOSIS study, a large multicenter, prospective observational study conducted in 14 countries, validated a single cut-off point of sFlt-1/PlGF ratio >38 in predicting preeclampsia [32]. However, less than 10% of the study population were Asian. Another research followed to validate the cut-off point (sFlt-1/PlGF ratio >38) among Asian women (China, Hong Kong, Japan, Singapore, South Korea, and Thailand) between December 2014 and December 2016 but Malaysia was not involved [46]. As sFlt-1 and PlGF levels may be influenced by ethnicity [47], there is a need to validate the test in Malaysia. Both researchers assessed the value of the sFlt-1/PlGF ratio for ruling out preeclampsia within a week and ruling in preeclampsia

**Table 2. Factors associated with preeclampsia.**

| Variable | Preeclampsia n (%) | Non preeclampsia n (%) | Crude OR | 95% CI | p | Adj. OR | 95% CI | p |
|---|---|---|---|---|---|---|---|---|
| **Age groups** | | | | | 0.139 | | | |
| More than 35 years old | 12(41.4%) | 63(56.8%) | 0.538 | (0.235;1.232) | | | | |
| ≤ 35 years old | 17(58.6%) | 48(43.2%) | 1.000 | | | | | |
| **Race** | | | | | | | | |
| Malay | 26(89.7%) | 88(79.3%) | 2.265 | (0.630; 0.149) | 0.177 | | | |
| Others | 3(10.3%) | 23(20.7%) | 1.000 | | | | | |
| **Occupational classification** | | | | | <0.001 | | | <0.001 |
| Professional | 15(51.7%) | 18(16.2%) | 5.536 | (2.282; 13.428) | | 6.694 | (2.247;19.937) | |
| Non-professional | 14(48.3%) | 93(83.8%) | 1.000 | | | 1.000 | | |
| **Household incomed class (median income)** | | | | | 0.266 | | | |
| Top 20 (> Ringgit Malaysia 10 960) | 4(13.8%) | 7(6.3%) | 3.214 | (0.761; 13.572) | 0.112 | | | |
| Middle 40 (Ringgit Malaysia 4850–10959) | 17(58.6%) | 59(53.2%) | 1.621 | (0.642; 4.090) | 0.306 | | | |
| Below 40 (< Ringgit Malaysia 4849) | 8(27.6%) | 45(40.5%) | 1.000 | | | | | |
| **Education level** | | | | | 0.541 | | | |
| Tertiary | 23(79.3%) | 82(73.9%) | 0.738 | (0.273; 1.992) | | | | |
| Secondary | 6(20.7%) | 29(26.1%) | 1.000 | | | | | |
| **Body mass index (BMI) class, kg/m$^2$** | | | | | | | | |
| BMI ≥25 | 22(75.9%) | 87(78.4%) | 0.867 | (0.331; 2.271) | 0.773 | | | |
| BMI ≤24.9 | 7(24.1%) | 24(21.6%) | 1.000 | | | | | |
| **Antenatal risk coded** | | | | | 0.445 | | | |
| Yellow (high risk) | 4(13.3%) | 22(19.8%) | 0.647 | (0.204; 2.052) | | | | |
| Green (medium risk) | 26(86.7%) | 89(80.2%) | 1.000 | | | | | |
| **Gravida classification** | | | | | 0.940 | | | |
| Gravida 1 | 6(20.7%) | 25(22.5%) | 1.200 | (0.117; 12.267) | 0.785 | | | |
| Gravida 2–5 | 22(75.9%) | 81(73.0%) | 1.358 | (0.151; 12.233) | 0.878 | | | |
| Gravida 6 and more | 1(3.4%) | 5(4.5%) | 1.000 | | | | | |
| **Parity classification** | | | | | 0.983 | | | |
| Multipara | 22(75.9%) | 84(75.7%) | 1.010 | (0.389;2.624) | | | | |
| Nulliparaous | 7(24.1%) | 27(24.3%) | 1.000 | | | | | |
| **POG at recruitment** | | | | | 0.482 | | | |
| 20–26 weeks | 3(10.3%) | 17(15.3%) | 0.638 | (0.174;2.346) | | | | |
| ≥27 weeks | 26(89.7%) | 94(84.7%) | 1.000 | | | | | |
| **Anaemia** | | | | | 0.335 | | | |
| Yes | 7(24.1%) | 18(16.2%) | 1.644 | (0.611; 4.420) | | | | |
| No | 22(75.9%) | 93(83.8%) | 1.000 | | | | | |
| **Diabetes mellitus** | | | | | 0.612 | | | |
| Yes | 2(6.9%) | 5(4.5%) | 1.570 | (0.289; 8.539) | | | | |
| No | 27(93.1%) | 106(95.5%) | 1.000 | | | | | |
| **Gestational DM** | | | | | 0.494 | | | |
| Yes | 9(31.0%) | 42(37.8%) | 0.739 | (0.308; 1.774) | | | | |
| No | 20(69.0%) | 69(62.2%) | 1.000 | | | | | |
| **Kidney disease** | | | | | 0.607 | | | |
| Yes | 1(3.4%) | 2(1.8%) | 1.946 | (0.170; 22.245) | | | | |

(*Continued*)

**Table 2.** (Continued)

| Variable | Preeclampsia n (%) | Non preeclampsia n (%) | Crude OR | 95% CI | p | Adj. OR | 95% CI | p |
|---|---|---|---|---|---|---|---|---|
| No | 28(96.6%) | 109(98.2%) | 1.000 | | | | | |
| **Autoimmune disease** | | | | | 0.607 | | | |
| Yes | 1(3.4%) | 2(1.8%) | 1.946 | (0.170; 22.245) | | | | |
| No | 28(96.6%) | 109(98.2%) | 1.000 | | | | | |
| **History of HDP** | | | | | 0.827 | | | |
| Yes | 2(6.9%) | 9(8.1%) | 0.840 | (0.171; 4.116) | | | | |
| No | 27(93.1%) | 102(91.9%) | 1.000 | | | | | |
| **Chronic hypertension** | | | | | 0.054 | | | |
| Yes | 7(24.1%) | 48(43.2%) | 0.418 | (0.165; 1.058) | | | | |
| No | 22(75.9%) | 63(56.8%) | 1.000 | | | | | |
| **sFlt-1/PlGF ratio** | | | | | <0.001 | | | <0.001 |
| Positive >38 | 17(58.6%) | 10(9.0%) | 14.308 | (5.349; 38.277) | | 15.063 | (5.206;43.557) | |
| Negative≤38 | 12(41.4%) | 101(91.0%) | 1.000 | | | 1.000 | | |

Final model using Backward Stepwise (Likelihood Ratio)

Hosmer and Lemeshow Test 0.343

Nagelkerke R Square 42.1%

Classification table 82.9%

within four weeks, with an established clinical value for the short-term prediction of pre-eclampsia in women with suspected preeclampsia, potentially helping to prevent unnecessary hospitalization and intervention. Our study evaluated the feasibility of the sFlt-1/PlGF ratio >38 to predict preeclampsia and adverse pregnancy outcome throughout pregnancy without limiting the period of outcome assessment.

A previous study on the same urban population in Kuala Lumpur revealed that women with gestational hypertension had significantly lower PlGF and higher sFlt-1 levels compared with normotensive women [48]. However, the study did not assess these biomarkers level among women with preeclampsia. Another study among 84 high risk Malaysian pregnant women who had at least one risk factor for preeclampsia observed a significantly higher median sFlt-1 and sFlt-1/PlGF ratio from 25 to 28 weeks of gestation and sFlt-1/PlGF ratio from 29 to 36 weeks in high risk women who developed preeclampsia [49]. The sFLT-1/PlGF ratio in the third trimester showed the best AUC of 87.3% (p<0.001, 95% CI 77.3–97.3) compared with second trimester AUC, which was 68.8% (p = 0.038, 95% CI 52.0–85.5). The optimized cut-off value for sFLT-1/PlGF ratio was 5.50 (sensitivity 92%; specificity 68%; PPV 32%; NPV 98%) with AUC 87.3% (95% CI, 77.3–97.3) [49]. However, the predictive value of these markers could not be clearly established due to a small samples size.

The current standard of antenatal care in maternal and child health clinics in Malaysia stratifies women at risk using clinical risk factors and a colour coding system. This has been proven to be insufficient [50]. Our finding shows that the antenatal color coding is not a significant predictor for preeclampsia. There is limited evidence on the effectiveness of routine blood pressure and urine protein screening to identify women with preeclampsia [10]. The percentage of women without proteinuria who developed preeclampsia or who have protein-uria without hypertension preceding preeclampsia is unclear due to different approaches in proteinuria measurement [10]. Presence of proteinuria is associated with a worse pregnancy outcome as compared to preeclampsia without proteinuria. On the contrary, higher levels of

**Table 3. Regression analysis between positive sFlt-1/PlGF ratio test and adverse pregnancy outcome.**

| sFlt-1/PlGF ratio | PREGNANCY OUTCOMES | | Crude OR[a] | 95% CI OR | p | Adjusted OR[b] | (95% CI) | p |
|---|---|---|---|---|---|---|---|---|
| | present | None | | | | | | |
| | n (%) | n (%) | | | | | | |
| **Maternal death** | | | | | | | | |
| Positive | 0(0%) | 27(19.3%) | - | - | - | | | |
| Negative | 0(0%) | 113(80.7%) | | | | | | |
| **Intubation** | | | | | 0.561 | | | |
| Positive | 1(33.3%) | 26(19.0%) | 2.135 | (0.186;4.445) | | | | |
| Negative | 2(66.7%) | 111(81.0%) | 1.000 | | | | | |
| **Caesarean section** | | | | | 0.386 | | | |
| Positive | 20(21.3%) | 7(15.2%) | 1.506 | (0.586;3.871) | | | | |
| Negative | 74(78.7%) | 39(84.8%) | 1.000 | | | | | |
| **Neonatal death** | | | | | 0.512 | | | |
| Positive | 0(0.0%) | 27(19.4%) | 0.000 | (0.000;—) | | | | |
| Negative | 1(100.0%) | 112(80.6%) | 1.000 | | | | | |
| **Low birth weight** | | | | | <0.001 | | | <0.001 |
| Positive | 10(52.6%) | 17(14.0%) | 6.797 | (2.412;19.160) | | 6.841 | (2.282;20.511) | |
| Negative | 9(47.4%) | 104(86.0%) | 1.000 | | | | | |
| **Premature delivery** | | | | | <0.001 | | | <0.001 |
| Positive | 16(48.5%) | 11(10.5%) | 8.043 | (3.188;20.289) | | 8.821 | (3.620;21.494) | |
| Negative | 17(51.5%) | 94(89.5%) | 1.000 | | | | | |
| **Apgar score at 5 minutes <7** | | | | | 0.001 | | | 0.028 |
| Positive | 11(84.6%) | 16(12.6%) | 38.156 | (7.741;188.084) | | 17.387 | (3.069;98.517) | |
| Negative | 2(15.4%) | 111(87.4%) | 1.000 | | | 1.000 | | |
| **DIVC** | | | | | - | | | |
| Positive | 0(0%) | 27(19.3%) | - | - | - | | | |
| Negative | 0(0%) | 113(80.7%) | - | | | | | |
| **Placenta abruptio** | | | | | - | | | |
| Positive | 1(100.0%) | 26(18.7%) | - | - | | | | |
| Negative | 0(0.0%) | 113(81.3%) | - | | | | | |
| **Acute heart failure** | | | | | - | | | |
| Positive | 0(0%) | 27(19.3%) | - | - | | | | |
| Negative | 0(0%) | 113(80.7%) | - | | | | | |
| **NICU admission** | | | | | | | | |
| Positive | 10(50.0%) | 17(14.2%) | 7.081 | (2.591; 19.352) | <0.001 | 8.305 | (2.815;24.501) | <0.001 |
| Negative | 10(50.0%) | 103(85.8%) | 1.000 | | | | | |
| **Intrauterine death** | | | | | | | | |
| Positive | 0(0%) | 27(19.3%) | - | - | | | | |
| Negative | 0(0%) | 113(80.7%) | - | | | | | |

[a] Crude odd ratio using simple logistic regression.

[b] Adjusted odd ratio (multiple logistic regression using Backward likelihood method, Hosmer and Lemenshow test p-value >0.05)

proteinuria among women diagnosed with preeclampsia does not indicate a higher risk of severe adverse outcomes [51]. The presence or absence of proteinuria is more important than the amount of proteinuria.

The progress of preeclampsia is often unpredictable and can lead to rapid deterioration in maternal and fetal conditions. Therefore, close surveillance is mandatory during antenatal visits [36]. A decision must be made at the time of diagnosis whether to manage the patient as an

**Table 4. The validity of sFLT-1/PlGF ratio test in predicting preeclampsia and low Apgar score at 5 minutes.**

| Outcome | Sensitivity | Specificity | PPV | NPV | LR + (95% CI) | LR- (95% CI) | cOR (95% CI) | aOR (95% CI) |
|---|---|---|---|---|---|---|---|---|
| Preeclampsia | 0.586 | 0.910 | 0.630 | 0.893 | 6.507 (3.344;12.660) | 0.455(0.293;0.704) | 14.308 (5.349;38.277) | 22.000 (6.448;75.069 |
| Apgar score at 5 minutes <7 | 0.846 | 0.874 | 0.407 | 0.982 | 6.716 (4.020;11.223) | 0.176(0.049;0.631) | 38.156 (7.741;188.084) | 17.387 (3.069;98.517) |
| Low birth weight | 0.526 | 0.860 | 0.370 | 0.920 | 3.746 (2.029;6.918) | 0.551(0.341;0.890) | 6.797 (2.412;19.160) | 6.841 (2.282;20.511) |
| Premature delivery | 0.485 | 0.895 | 0.593 | 0.847 | 4.628 (2.391; 8.959) | 0.575(0.411;0.806) | 8.043 (3.188;20.289) | 8.821 (3.620;21.494) |
| NICU admission | 0.500 | 0.858 | 0.370 | 0.912 | 3.529 (1.896;6.570) | 0.583(0.374;0.909) | 7.081 (2.591; 19.352) | 8.305 (2.815;24.501) |

outpatient or inpatient [36]. In the absence of proteinuria, the mother may be managed as an outpatient [36]. However, it is now recognized that preeclampsia may be present even without proteinuria. A highly predictive, pre-emptive, and low-cost diagnostic test is needed to guide management [52]. Early preeclampsia detection through universal or targeted screening methods may help reduce health related adverse events, particularly for newborns [53]. We found positive sFlt-1/PlGF ratio (>38) to have 15 times higher odds of developing preeclampsia and 6 to 17 times higher odds of having a newborn with poor outcomes. The majority of positive

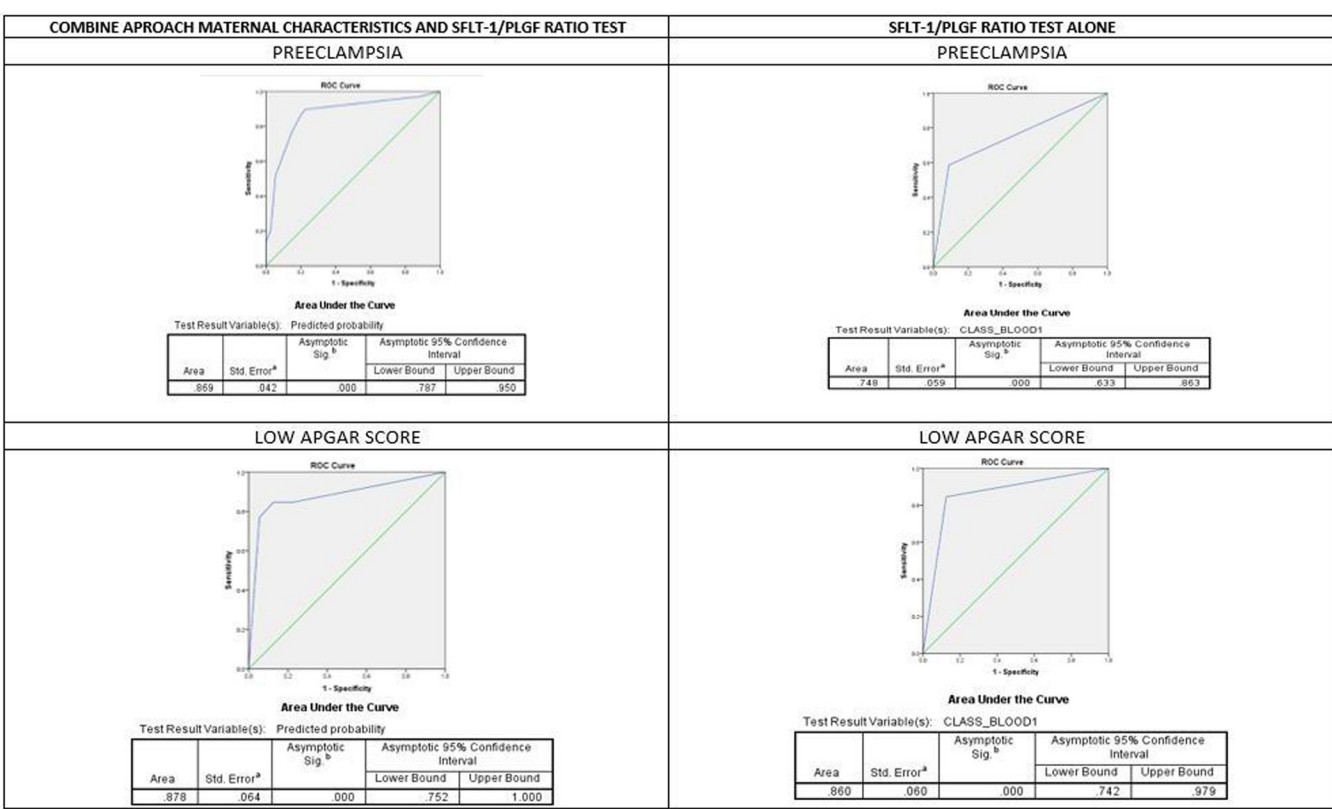

**Fig 3. ROC curve comparison between combine maternal and biomarker approach and biomarker alone.**

test mothers (62.5%) manifest preeclampsia within a week. With good NPV (89.3%) it is effective in identifying patients who will benefit from admission, close monitoring, and antenatal corticosteroids commencement in anticipation of preterm delivery [54, 55]. Thus, the test may improve maternal and child health care by early identification, early intervention and prevent subsequent bad outcomes.

Placental dysfunction causes a range of perinatal pathologies, including preeclampsia. Angiogenesis-related factors, including sFlt-1 and PlGF, play an important role in placental dysfunction. Altered levels are detectable several weeks prior to the onset of pregnancy complications [11, 56]. We found that, among mothers who developed preeclampsia within a week of assessment, 62% (5/8) had sFLT-1/PlGF ratio >38 (positive). A total of 59% (17/29) women with such positive results developed preeclampsia within a week, and majority (94%) developed preeclampsia within four weeks. However, 37.9% women who developed preeclampsia were among women with sFlt-1/PlGF ratio ≤38 (negative) with a small percentage of 25% (3/12) progressing to preeclampsia in less than a week of assessment. Besides the ability to predict preeclampsia, the positive test can predict Apgar score < 7 at 5 minutes postnatally with a sensitivity of 84.6%, specificity of 87.4%, PPV of 40.7% and NPV of 98.2%. The AUC was 86.0% (p<0.001, 95% CI 74.2–97.9) using sFlt-1/PlGF ratio alone. The AUC and predictive values showed improvement based on multiple logistic regression predictive values.

An interesting finding in this study was that women working as professionals were at higher risk of developing preeclampsia with adjusted odds ratio (aOR) 8.79 (95% CI 2.80–27.58). This finding is supported by a previous study that the risk of preeclampsia increased 3.1-fold (95% CI 1.2–7.8) among women employed in high-stress jobs (high psychological demand, low decision latitude) [57]. The aOR was 2.0 (95% CI 1.0–4.3) for low-stress jobs compared to non-working women [57]. In another study, being exposed to physically demanding and stressful occupational conditions at the onset of pregnancy increased the risk of preeclampsia [58]. Women working more than five successive days without a day off had aOR 3.0 (95% CI 1.0–9.5) [58]. A significant association exists between acute and chronic occupational/domestic stress with major depression/current depressive symptoms among married professionals and managerial employees [59]. We therefore propose adding occupation as a criteria in the list of current guidelines for risk assessment of preeclampsia in order to improve preeclampsia prediction and thus prevention.

The sFlt-1/PlGF ratio is useful for clinical decision-making with regard to hospitalization [60]. In addition to improving clinical care, assessment of the sFlt-1/PlGF ratio helps avoid unnecessary stress and anxiety for the patient [54]. Malaysia practices a mixed public-private healthcare system [61]. Nonetheless, public hospitals and clinics are the primary source of care in Malaysia. The public sector provides for the bulk (65%) of the population, but is served by only 45% of all registered doctors, and even fewer specialists (25–30%) [62]. The heavily subsidized public sector is borne by budget provisions, with patients paying minimal fees for access to both outpatient and hospitalized care [61, 62]. Thus, a reduction in hospitalization may contribute substantially to cost-saving, and reduce the financial and workload burden in the Malaysian healthcare system [63–65].

The strength of our study lies in the prospective cohort study design, hence the temporal effect between the sFlt:PlGF and onset of preeclampsia and adverse pregnancy outcome can be established according to Hills criteria. The limitation of this study is that it was restricted to a single urban tertiary centre, hence producing low external validity. A number of study participants (43%) were lost to follow-up especially after the first test due to several reasons, including government enforcement of the movement control order in response to the COVID-19 pandemic, work and family commitments, and logistic issues. Therefore, the findings should be interpreted cautiously if it were to be applied to a different population setting.

## Conclusions

We found a higher PPV value for sFlt-1/PlGF compared to previous observations. Interestingly, we also found that the sFlt-1/PlGF ratio is a significant marker for predicting adverse pregnancy outcomes among newborns of medium to high risk mothers. In summary, this study highlighted for the first time, the utility of the sFlt-1/PlGF ratio as a feasible biomarker to predict preeclampsia in the Malaysian population, in combination with other parameters (maternal clinical characteristics), provided the outcome measurement period is not restricted to 4 weeks.

## Supporting information

**S1 Data.**
(XLSX)

## Acknowledgments

The authors would like to acknowledge the contribution of all clinicians, scientists, technologists, and patients who were involved in this research.

## Author Contributions

**Conceptualization:** Nurul Afzan Aminuddin, Zaleha Abdullah Mahdy, Rahana Abd Rahman, Dian Nasriana Nasuruddin.

**Data curation:** Nurul Afzan Aminuddin, Rosnah Sutan, Rahana Abd Rahman, Dian Nasriana Nasuruddin.

**Formal analysis:** Nurul Afzan Aminuddin, Rosnah Sutan.

**Funding acquisition:** Nurul Afzan Aminuddin, Rosnah Sutan, Zaleha Abdullah Mahdy, Rahana Abd Rahman.

**Investigation:** Nurul Afzan Aminuddin, Rosnah Sutan, Zaleha Abdullah Mahdy, Rahana Abd Rahman.

**Methodology:** Nurul Afzan Aminuddin, Rosnah Sutan, Zaleha Abdullah Mahdy, Rahana Abd Rahman, Dian Nasriana Nasuruddin.

**Project administration:** Nurul Afzan Aminuddin, Rosnah Sutan, Zaleha Abdullah Mahdy, Dian Nasriana Nasuruddin.

**Resources:** Nurul Afzan Aminuddin, Rosnah Sutan, Zaleha Abdullah Mahdy, Dian Nasriana Nasuruddin.

**Software:** Dian Nasriana Nasuruddin.

**Supervision:** Rosnah Sutan, Zaleha Abdullah Mahdy, Rahana Abd Rahman.

**Validation:** Rosnah Sutan, Zaleha Abdullah Mahdy.

**Writing – original draft:** Nurul Afzan Aminuddin.

**Writing – review & editing:** Rosnah Sutan, Zaleha Abdullah Mahdy, Rahana Abd Rahman, Dian Nasriana Nasuruddin.

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
