## [Decision Letter · Decision Letter 0]

13 Sep 2021

PONE-D-21-25053The feasibility of Soluble Fms-Like Tyrosine Kinase-1 (sFLT-1) and Placental Growth Factor (PlGF) ratio biomarker in predicting preeclampsia and adverse pregnancy outcomes among medium to high risk mothersPLOS ONE

Dear Dr. sutan,

Thank you for submitting your manuscript to PLOS ONE. After careful consideration, we feel that it has merit but does not fully meet PLOS ONE’s publication criteria as it currently stands. Therefore, we invite you to submit a revised version of the manuscript that addresses the points raised during the review process.

We look forward to receiving your revised manuscript.

Kind regards,

Muhammad Tarek Abdel Ghafar, M.D

Academic Editor

PLOS ONE

Journal Requirements:

2. During your revisions, please note that a simple title correction is required: please add the location at which the study has been undertaken (i.e. The feasibility of Soluble Fms-Like Tyrosine Kinase-1 (sFLT-1) and Placental Growth Factor (PlGF) ratio biomarker in predicting preeclampsia and adverse pregnancy outcomes among medium to high risk mothers in Kuala Lumpur, Malaysia). Please ensure this is updated in the manuscript file and the online submission information.

[Funding

This research received funding through the Fundamental Grant (FF-2019-371) and Matching Grant (FF-2019-371/1) from Universiti Kebangsaan Malaysia.

Conflict of Interest

Provision of Elecsys sFlt-1/PlGF and PreciControl Multimaker test kits from Roche Diagnostics International Ltd is acknowledged.

The funders had no role in study design, data collection and analysis, decision to publish, or preparation of the manuscript.]

4. Please ensure that you include a title page within your main document. You should list all authors and all affiliations as per our author instructions and clearly indicate the corresponding author.

Reviewers' comments:

Reviewer's Responses to Questions

**Comments to the Author**

1. Is the manuscript technically sound, and do the data support the conclusions?

Reviewer #1: Yes

Reviewer #2: Yes

Reviewer #3: Yes

2. Has the statistical analysis been performed appropriately and rigorously? 

Reviewer #1: Yes

Reviewer #2: Yes

Reviewer #3: Yes

3. Have the authors made all data underlying the findings in their manuscript fully available?

Reviewer #1: Yes

Reviewer #2: Yes

Reviewer #3: Yes

4. Is the manuscript presented in an intelligible fashion and written in standard English?

Reviewer #1: Yes

Reviewer #2: Yes

Reviewer #3: Yes

5. Review Comments to the Author

Reviewer #1: Generally, the design of this study is well attempted and these results must be helpful to the pregnant women in Malaysia. Interestingly, the occupation may impact on the development of preeclampsia in Malaysian.

Major revision:

In the algorithm of this study, some population may change the results of sFLT-1/PlGF ratio, positive to negative, but clinically the development of preeclampsia seems ‘one way’. If some rare cases exist in this study, please tell the number and the process or outcome of them.

Minor revision:

#1. In Table 1, according to the sentence, line 157, you should correct the small title of ‘History of PIH’ to ‘History of HDP’.

#2. In Fig. 3 you showed the wrong number in the major title. Not Fig. 2 but Fig. 3.

#3. In Table 4, you also have to correct the title, ‘1 minute’ to ‘5 minutes’.

Reviewer #2: 1. Summary of Research

In this paper, the authors evaluated the feasibility of the sFLT-1/PlGF biomarker ratio in predicting preeclampsia and adverse pregnancy outcomes using a single cut-off point of >38 among mothers in Malaysia. The study confirms that although sFLT-1/P1GF ratio is a feasible biomarker for predicting pregnancy, a combined approach with maternal clinical characteristics provides a better prediction for preeclampsia and adverse pregnancy outcomes.

Below are some comments for the authors:

2. Specific Areas

Introduction

Language editing is recommended at line 41 of the manuscript.

Materials and Methods:

Authors should indicate the full meaning of the abbreviation, ISSHP, at the first mention in the write up at line 113 of the manuscript.

Results and Discussion

Authors indicated that 57% of the study participants completed the research algorithm. Were there reasons or factors that explain why a significant number were lost to follow up especially after the first test?

3. Additional Comments

Authors should kindly be consistent with referencing style and not mix Harvard with Vancouver as identified at line 293-294 and 310.

Reviewer #3: The author analyzes the relationship between the sFlt-1 / PlGF ratio and preeclampsia in detail. The content is very interesting and orderly. But I'm not sure what's new.　That is the reason I don't think it is suitable for this journal.　This study is very interesting, so I would be happy if you rewrite the conclusions so that you can easily understand what is new.

6. PLOS authors have the option to publish the peer review history of their article (what does this mean?). If published, this will include your full peer review and any attached files.

Reviewer #1: No

Reviewer #2: **Yes: **Dr. Timothy Kwabena Adjei

Reviewer #3: No

---

## [Author Response · Author response to Decision Letter 0]

29 Sep 2021

EDITOR SUGGESTIONS:

Journal Requirements:

• RESPONSE: File naming was edited to comply with the style requirements. We hope it is now correct and complies with the style requirements.

2. During your revisions, please note that a simple title correction is required: please add the location at which the study has been undertaken (i.e. The feasibility of Soluble Fms-Like Tyrosine Kinase-1 (sFLT-1) and Placental Growth Factor (PlGF) ratio biomarker in predicting preeclampsia and adverse pregnancy outcomes among medium to high risk mothers in Kuala Lumpur, Malaysia). Please ensure this is updated in the manuscript file and the online submission information.

• RESPONSE: Thank you for providing these insights. We have corrected the manuscript title as per suggestion and updated in the manuscript file and the online submission information.

Funding

This research received funding through the Fundamental Grant (FF-2019-371) and Matching Grant (FF-2019-371/1) from Universiti Kebangsaan Malaysia.

Conflict of Interest

Provision of Elecsys sFlt-1/PlGF and PreciControl Multimaker test kits from Roche Diagnostics International Ltd is acknowledged.

• RESPONSE : We have included the following statement "This does not alter our adherence to PLOS ONE policies on sharing data and materials” as an updated Competing Interests statement in the cover letter

4. Please ensure that you include a title page within your main document. You should list all authors and all affiliations as per our author instructions and clearly indicate the corresponding author.

• RESPONSE: We have included the title page within the main document. We have listed all authors and all affiliations as per the author instructions and indicated the corresponding author.

• RESPONSE : We have included a separate caption for each figure in the main manuscript body

REVIEWER #1 COMMENTS:

Generally, the design of this study is well attempted and these results must be helpful to the pregnant women in Malaysia. Interestingly, the occupation may impact on the development of preeclampsia in Malaysian.

6. In the algorithm of this study, some population may change the results of sFLT-1/PlGF ratio, positive to negative, but clinically the development of preeclampsia seems ‘one way’. If some rare cases exist in this study, please tell the number and the process or outcome of them.

• RESPONSE : That is an interesting query. However, there were no positive case change to negative result in our sample population. 

7. In Table 2, according to the sentence, line 157, you should correct the small title of ‘History of PIH’ to ‘History of HDP’.

• RESPONSE: Thank you for this observation. I have corrected the small title of ‘History of PIH’ to ‘History of HDP’ in Table 2 in the revised manuscript.

8. In Fig. 3 you showed the wrong number in the major title. Not Fig. 2 but Fig. 3.

• RESPONSE : Our apologies for the mistake. We have included a new Figure 3 with the correct title. 

9. In Table 4, you also have to correct the title, “1 minute’ to ‘5 minutes”

• RESPONSE: Thank you for this observation. I have corrected the title, “1 minute’ to ‘5 minutes” in Table 4.

REVIEWER #2 COMMENTS:

In this paper, the authors evaluated the feasibility of the sFLT-1/PlGF biomarker ratio in predicting preeclampsia and adverse pregnancy outcomes using a single cut-off point of >38 among mothers in Malaysia. The study confirms that although sFLT-1/PlGF ratio is a feasible biomarker for predicting pregnancy, a combined approach with maternal clinical characteristics provides a better prediction for preeclampsia and adverse pregnancy outcomes.

10. Specific Areas:

Introduction

Language editing is recommended at line 41 of the manuscript.

• RESPONSE: We agree with your assessment. Language editing has been performed as in lines 57-59 in the revised manuscript, as follows: “Identifying women at risk of developing preeclampsia for early intervention with commencement of aspirin/calcium as prophylaxis, close monitoring, and timely delivery are critical steps in the management to prevent adverse pregnancy outcomes.”.

Materials and Methods:

Authors should indicate the full meaning of the abbreviation, ISSHP, at the first mention in the write up at line 113 of the manuscript.

• RESPONSE: Thank you for this suggestion. We have stated in full the meaning of the abbreviation ISSHP on page 7, lines 134-135, as “International Society for the Study of Hypertension in Pregnancy”.

Results and Discussion

Authors indicated that 57% of the study participants completed the research algorithm. Were there reasons or factors that explain why a significant number were lost to follow up especially after the first test?

• RESPONSE: You have raised an important point. There are several reasons to explain why a significant number of study participants were lost to follow up especially after the first test, including the government enforcement of the movement control order in response to the COVID-19 pandemic, work and family commitments, and logistic issues. We have included this point in the Discussion Section in Lines 363-366.

Authors should kindly be consistent with referencing style and not mix Harvard with Vancouver as identified at line 293-294 and 310.

• RESPONSE: We has edited the referencing style to Vancouver format as in lines 320-321 and 339.

REVIEWER #3 COMMENTS: 

The author analyzes the relationship between the sFlt-1 / PlGF ratio and preeclampsia in detail. The content is very interesting and orderly. But I'm not sure what's new.　That is the reason I don't think it is suitable for this journal.　This study is very interesting, so I would be happy if you rewrite the conclusions so that you can easily understand what is new.

• RESPONSE : Thank you for providing these useful insights. We has rewritten the conclusions and highlighted the new perspectives of this study as in lines 45-48, 278-280 and 369-373.

---

## [Decision Letter · Decision Letter 1]

19 Oct 2021

PONE-D-21-25053R1The feasibility of Soluble Fms-Like Tyrosine Kinase-1 (sFLT-1) and Placental Growth Factor (PlGF) ratio biomarker in predicting preeclampsia and adverse pregnancy outcomes among medium to high risk mothers in Kuala Lumpur, MalaysiaPLOS ONE

Dear Dr. sutan,

Thank you for submitting your manuscript to PLOS ONE. After careful consideration, we feel that it has merit but does not fully meet PLOS ONE’s publication criteria as it currently stands. Therefore, we invite you to submit a revised version of the manuscript that addresses the points raised during the review process. Please submit your revised manuscript by Dec 03 2021 11:59PM. If you will need more time than this to complete your revisions, please reply to this message or contact the journal office at plosone@plos.org. Please include the following items when submitting your revised manuscript:A rebuttal letter that responds to each point raised by the academic editor and reviewer(s). You should upload this letter as a separate file labeled 'Response to Reviewers'.A marked-up copy of your manuscript that highlights changes made to the original version. You should upload this as a separate file labeled 'Revised Manuscript with Track Changes'.An unmarked version of your revised paper without tracked changes. You should upload this as a separate file labeled 'Manuscript'.If applicable, we recommend that you deposit your laboratory protocols in protocols.io to enhance the reproducibility of your results. Protocols.io assigns your protocol its own identifier (DOI) so that it can be cited independently in the future. For instructions see: https://journals.plos.org/plosone/s/submission-guidelines#loc-laboratory-protocols. Additionally, PLOS ONE offers an option for publishing peer-reviewed Lab Protocol articles, which describe protocols hosted on protocols.io. Read more information on sharing protocols at https://plos.org/protocols?utm_medium=editorial-email&utm_source=authorletters&utm_campaign=protocols.

We look forward to receiving your revised manuscript.

Kind regards,

Muhammad Tarek Abdel Ghafar, M.D

Academic Editor

PLOS ONE

Journal Requirements:

Reviewers' comments:

Reviewer's Responses to Questions

**Comments to the Author**

1. If the authors have adequately addressed your comments raised in a previous round of review and you feel that this manuscript is now acceptable for publication, you may indicate that here to bypass the “Comments to the Author” section, enter your conflict of interest statement in the “Confidential to Editor” section, and submit your "Accept" recommendation.

Reviewer #1: All comments have been addressed

Reviewer #2: All comments have been addressed

2. Is the manuscript technically sound, and do the data support the conclusions?

Reviewer #1: Yes

Reviewer #2: Yes

3. Has the statistical analysis been performed appropriately and rigorously? 

Reviewer #1: Yes

Reviewer #2: Yes

4. Have the authors made all data underlying the findings in their manuscript fully available?

Reviewer #1: Yes

Reviewer #2: Yes

5. Is the manuscript presented in an intelligible fashion and written in standard English?

Reviewer #1: Yes

Reviewer #2: Yes

6. Review Comments to the Author

Reviewer #1: This revised manuscript has been fully improved in response to our previous suggestions. I have no further questions or queries in my head.

Thank you for your brief and accurate jobs.

Reviewer #2: Initial comments have all been addressed by authors.

Below are few additional comments for authors to address.

1. Authors should kindly review the definition of preeclampsia to be “Hypertension with proteinuria and/or maternal organ damage…” rather than 'or' in the introduction and the entire write-up.

2. Per the Aspre trial by FMF group, optimal benefits of low dose aspirin are between 12-16 gestational weeks. It will be important to know from the authors if women who were high risk for preeclampsia (chronic hypertension, previous HDP, etc) were on aspirin before being recruited into the study. If not, were they started after 20 gestational weeks? This will be essential to know because it will be unethical or controversial to delay onset of aspirin in such high risk group. Is the practice of administering LD aspirin to such high risk women incorporated in the Malaysian Clinical Practice Guidelines (CPG)?

7. PLOS authors have the option to publish the peer review history of their article (what does this mean?). If published, this will include your full peer review and any attached files.

Reviewer #1: No

Reviewer #2: **Yes: **Dr. Timothy K. Adjei

---

## [Author Response · Author response to Decision Letter 1]

18 Nov 2021

EDITOR SUGGESTIONS:

Journal Requirements:

1. Please review your reference list to ensure that it is complete and correct. If you have cited papers that have been retracted, please include the rationale for doing so in the manuscript text or remove these references and replace them with relevant current references. Any changes to the reference list should be mentioned in the rebuttal letter that accompanies your revised manuscript. If you need to cite a retracted article, indicate the article’s retracted status in the References list and also include a citation and full reference for the retraction notice.

• RESPONSE: We have reviewed the references. Therefore, we made some changes as stated below.

(i) There are few duplicate references:

(1) References number 60 and 61. We have deleted reference number 61 and replaced it with citation number 60.

(2) References number 58 and 59. We have deleted reference number 59 and replaced it with citation number 58.

(3) References number 32 and 45. We have deleted reference number 45 and replaced it with citation number 32.

(4) We renumbered again to make it align in both manuscript body and in the reference list.

(ii) We don’t cite any papers that have been retracted.

REVIEWER #2 COMMENTS:

Reviewer #2: Initial comments have all been addressed by authors.

Below are few additional comments for authors to address.

1. Authors should kindly review the definition of preeclampsia to be “Hypertension with proteinuria and/or maternal organ damage…” rather than 'or' in the introduction and the entire write-up.

• RESPONSE: We agree with your assessment. We have changed the definition as below:

line 52 page 3 “Preeclampsia is characterized by the combined presentation of hypertension with proteinuria and/or maternal organ dysfunction, such as renal insufficiency, liver involvement, neurological or haematological complications, and uteroplacental dysfunction as evidenced by fetal growth restriction (FGR) (2, 6).”

Line 131 page 7 – ‘hypertension with proteinuria and/or maternal organ dysfunction’

2. Per the Aspre trial by FMF group, optimal benefits of low dose aspirin are between 12-16 gestational weeks. It will be important to know from the authors if women who were high risk for preeclampsia (chronic hypertension, previous HDP, etc) were on aspirin before being recruited into the study. If not, were they started after 20 gestational weeks? This will be essential to know because it will be unethical or controversial to delay onset of aspirin in such high risk group. Is the practice of administering LD aspirin to such high risk women incorporated in the Malaysian Clinical Practice Guidelines (CPG)?

RESPONSE: You have raised an important point. Women who were high risk for preeclampsia were on aspirin before being recruited into this study. Our enrolment of respondents into the study starts at �20 weeks’ gestation until delivery (Line 105, page 6). The Malaysian Clinical Practice Guidelines (CPG) did incorporated practice of administering low dose aspirin to high-risk women for preeclampsia such as:

• Hypertensive disease during previous pregnancy

• Chronic kidney disease

• Autoimmune disease such as Systemic Lupus Erythematosus (SLE) or anti-phospholipid syndrome (APS)

• Type 1 or type 2 diabetes mellitus

• Chronic hypertension

We did not delay or alter any management as written in the CPG.

Figure 1, figure 2 and figure have been save in TIF format and uploaded

---

## [Editor Report · Decision Letter 2]

24 Nov 2021

PONE-D-21-25053R2The feasibility of Soluble Fms-Like Tyrosine Kinase-1 (sFlt-1) and Placental Growth Factor (PlGF) ratio biomarker in predicting preeclampsia and adverse pregnancy outcomes among medium to high risk mothers in Kuala Lumpur, MalaysiaPLOS ONE

Dear Dr. sutan,

Thank you for submitting your manuscript to PLOS ONE. After careful consideration, we feel that it has merit but does not fully meet PLOS ONE’s publication criteria as it currently stands. Therefore, we invite you to submit a revised version of the manuscript that addresses the points raised during the review process.

We look forward to receiving your revised manuscript.

Kind regards,

Muhammad Tarek Abdel Ghafar, M.D

Academic Editor

PLOS ONE

Journal Requirements:

Additional Editor Comments:

We only need the revised manuscript (clean copy) and revised manuscript with track changes and point-to-point response to the reviewers' recent comments. There are some duplicate files belonging to previous revisions. Please delete any redundant files from previous reviews when you resubmit the revision.
---

## [Author Response · Author response to Decision Letter 2]

6 Dec 2021

Journal Requirements:

Please review your reference list to ensure that it is complete and correct. If you have cited papers that have been retracted, please include the rationale for doing so in the manuscript text, or remove these references and replace them with relevant current references. Any changes to the reference list should be mentioned in the rebuttal letter that accompanies your revised manuscript. If you need to cite a retracted article, indicate the article has retracted status in the References list and also include a citation and full reference for the retraction notice. 

RESPONSE: We have reviewed the references. Therefore, we made some changes as stated below.

There are a few duplicate references:

(1) References number 60 and 61. We have deleted reference number 61 and replaced it with citation number 60.

(2) References number 58 and 59. We have deleted reference number 59 and replaced it with citation number 58.

(3) Corrected the references cited as numbers 32 and 45. We have deleted reference number 45 and replaced it with citation number 32.

(4) We change the sequence number of reference 39 become 40 and 40 become 39 as following the text in the manuscript.

(5) We renumbered again to make it align in both the manuscript body and in the reference list.

We do not cite any papers that have been retracted.

Additional Editor Comments:

We only need the revised manuscript (clean copy) and revised manuscript with track changes and point-to-point response to the reviewers' recent comments. There are some duplicate files belonging to previous revisions. Please delete any redundant files from previous reviews when you resubmit the revision.

RESPONSE: we resubmitted the revised manuscript(clean copy). We labelled as manuscript_5th Dec21, and a revised manuscript with track changes as Revised Manuscript with Track Changes_5Dec21. We respond point-to-point response to the reviewers' recent comments. Duplication of previous revisions has been deleted in this presence resubmission version.

REVIEWER #2 COMMENTS:

few additional comments for authors to address.

1. Authors should kindly review the definition of preeclampsia to be "Hypertension with proteinuria and/or maternal organ damage…" rather than 'or' in the introduction and the entire write-up.

• RESPONSE: We agree with your assessment. We have changed the definition as below:

line 52 page 3 "Preeclampsia is characterized by the combined presentation of hypertension with proteinuria and/or maternal organ dysfunction, such as renal insufficiency, liver involvement, neurological or haematological complications, and uteroplacental dysfunction as evidenced by fetal growth restriction (FGR) (2, 6)."

Line 131 page 7 – 'hypertension with proteinuria and/or maternal organ dysfunction'

2. Per the Aspre trial by FMF group, optimal benefits of low dose aspirin are between 12-16 gestational weeks. It will be important to know from the authors if women who were high risk for preeclampsia (chronic hypertension, previous HDP, etc) were on aspirin before being recruited into the study. If not, were they started after 20 gestational weeks? This will be essential to know because it will be unethical or controversial to delay onset of aspirin in such high risk group. Is the practice of administering LD aspirin to such high risk women incorporated in the Malaysian Clinical Practice Guidelines (CPG)?

RESPONSE: You have raised an important point. Women who were high risk for preeclampsia were on aspirin before being recruited into this study. Our enrolment of respondents into the study starts at �20 weeks' gestation until delivery (Line 105, page 6). The inclusion criteria for enrollment to study is stated (line 115-118). The Malaysian Clinical Practice Guidelines (CPG) did the incorporated practice of administering low dose aspirin to high-risk women for preeclampsia such as:

• Hypertensive disease during a previous pregnancy

• Chronic kidney disease

• Autoimmune disease such as Systemic Lupus Erythematosus (SLE) or anti-phospholipid syndrome (APS)

• Type 1 or type 2 diabetes mellitus

• Chronic hypertension

We did not delay or alter any management as written in the CPG.

Figure 1, figure 2 and figure have been saved in TIF format and uploaded

---

## [Decision Letter · Decision Letter 3]

20 Jan 2022

PONE-D-21-25053R3The feasibility of Soluble Fms-Like Tyrosine Kinase-1 (sFlt-1) and Placental Growth Factor (PlGF) ratio biomarker in predicting preeclampsia and adverse pregnancy outcomes among medium to high risk mothers in Kuala Lumpur, MalaysiaPLOS ONE

Dear Dr. sutan,

Thank you for submitting your manuscript to PLOS ONE. After careful consideration, we feel that it has merit but does not fully meet PLOS ONE’s publication criteria as it currently stands. Therefore, we invite you to submit a revised version of the manuscript that addresses the points raised during the review process.

We look forward to receiving your revised manuscript.

Kind regards,

Muhammad Tarek Abdel Ghafar, M.D

Academic Editor

PLOS ONE

Additional Editor Comments:

1- Table 2: Gravida classification is not significant in univariate analysis, so it should not be included in multivariate analysis.

2- Table 3: Adjusted OR of Low birth weight, premature delivery, and NICU admission should be added.

3- Abbreviations such as PPV and NPV should be defined in full-term before their first mention in the abstract.

Reviewers' comments:

Reviewer's Responses to Questions

**Comments to the Author**

1. If the authors have adequately addressed your comments raised in a previous round of review and you feel that this manuscript is now acceptable for publication, you may indicate that here to bypass the “Comments to the Author” section, enter your conflict of interest statement in the “Confidential to Editor” section, and submit your "Accept" recommendation.

Reviewer #2: All comments have been addressed

2. Is the manuscript technically sound, and do the data support the conclusions?

Reviewer #2: Yes

3. Has the statistical analysis been performed appropriately and rigorously? 

Reviewer #2: Yes

4. Have the authors made all data underlying the findings in their manuscript fully available?

Reviewer #2: Yes

5. Is the manuscript presented in an intelligible fashion and written in standard English?

Reviewer #2: Yes

6. Review Comments to the Author

Reviewer #2: The Authors have addressed all comments and concerns raised to my satisfaction. I have no additional comments.

7. PLOS authors have the option to publish the peer review history of their article (what does this mean?). If published, this will include your full peer review and any attached files.

Reviewer #2: **Yes: **Dr. Timothy Kwabena Adjei

---

## [Author Response · Author response to Decision Letter 3]

29 Jan 2022

thank you for the valuable comments. here is our response according to the each comments given. 

1. Table 2: Gravida classification is not significant in univariate analysis, so it should not be included in multivariate analysis.

• RESPONSE: We have removed gravida classification in multivariate analysis and reanalyzed. The new result from multivariate analysis was updated in Table 2 (page 12,line 222, 236-238), Abstract(page 3, line 40 to 45, 48-49, 51-52) and in discussion part(page 20, line 335-337)

2. Table 3: Adjusted OR of Low birth weight, premature delivery, and NICU admission should be added.

• RESPONSE: We have added Adjusted OR of Low birth weight, premature delivery, and NICU admission in Table 3 (page 15-16, line 253) and Table 4(page 17, line 276) and description at page 14 line 250

3. Abbreviations such as PPV and NPV should be defined in full-term before their first mention in the abstract.

• RESPONSE: We have defined full-term of PPV and NPV abbreviations before their first mention in the abstract (page 3).

---

## [Editor Report · Decision Letter 4]

23 Feb 2022

The feasibility of Soluble Fms-Like Tyrosine Kinase-1 (sFlt-1) and Placental Growth Factor (PlGF) ratio biomarker in predicting preeclampsia and adverse pregnancy outcomes among medium to high risk mothers in Kuala Lumpur, Malaysia

PONE-D-21-25053R4

Dear Dr. sutan,

We’re pleased to inform you that your manuscript has been judged scientifically suitable for publication and will be formally accepted for publication once it meets all outstanding technical requirements.

Kind regards,

Muhammad Tarek Abdel Ghafar, M.D

Academic Editor

PLOS ONE
---

## [Editor Report · Acceptance letter]

2 Mar 2022

PONE-D-21-25053R4 

The feasibility of Soluble Fms-Like Tyrosine Kinase-1 (sFLT-1) and Placental Growth Factor (PlGF) ratio biomarker in predicting preeclampsia and adverse pregnancy outcomes among medium to high risk mothers in Kuala Lumpur, Malaysia 

Dear Dr. Sutan:

I'm pleased to inform you that your manuscript has been deemed suitable for publication in PLOS ONE. Congratulations! Your manuscript is now with our production department. 

Kind regards, 

on behalf of

Prof Muhammad Tarek Abdel Ghafar 

Academic Editor

PLOS ONE